# Diagnostic accuracy of the ESAT6-CFP10 skin test for latent tuberculosis infection among jail detainees

Xinru Fei,[1,2] Shanshan Wang,[3] Zhan Wang,[1,2] Xinsong Hu,[1,2] Cheng Chen,[1,2] Limei Zhu,[1,2] Leonardo Martinez,[4] Peijun Tang,[5] Qiao Liu[1,2]

**ABSTRACT**  As an alternative to the tuberculin skin test (TST) or QuantiFERON-TB Gold In-Tube (QFT-GIT), ESAT6-CFP10 (EC) skin test is an emerging screening method; however, its value in the diagnosis of latent tuberculosis infection (LTBI) in detainees is still unclear in China. Newly admitted detainees meeting inclusion criteria were enrolled, with demographic/clinical data collected via structured questionnaires. TST, EC skin test, and QFT-GIT screenings were performed, documenting the diameter of skin indurations and/or redness at injection sites and blistering reactions at injection sites. In total, 1,038 detainees were enrolled in this study from October 2022 to October 2023 with 236 LTBI (22.7%). The positive rate of TST, EC skin test, and QFT-GIT was 18.1%, 10.6% and 11.9%. The area under the curve for EC was 0.820, indicating a strong concordance with QFT-GIT ($\kappa$ = 0.673). Compared with QFT-GIT, the sensitivity of EC was 66.9%, and the specificity was 97.0%. The mean induration diameter or redness of EC was significantly larger than that of TST ($P < 0.001$). In the regression model, no history of alcohol consumption (aOR = 0.433, 95% confidence interval [CI]: 0.200, 0.938), no history of surgical trauma (aOR = 0.731, 95% CI: 0.539, 0.991), and no drug use (aOR = 0.473, 95% CI: 0.233, 0.961) was identified as a protective factor for LTBI. The EC demonstrated both high specificity and sensitivity comparable to the QFT-GIT. When screening for LTBI among jail detainees in this setting, particular attention should be given to individuals with a history of alcohol consumption, surgical trauma, and drug use.

**IMPORTANCE**  Jail detainees represent a vulnerable population with an elevated risk of tuberculosis. The EC skin test demonstrates promising potential as an alternative to traditional diagnostic methods, such as the TST and QFT-GIT assay, for LTBI screening. Targeted screening strategies can facilitate the early detection, diagnosis, and management of LTBI.

**KEYWORDS**    tuberculosis, ESAT6-CFP10, tuberculosis infection, diagnostic value

Tuberculosis (TB) is the top infectious disease killer globally. An estimated 10.6 million persons developed tuberculosis worldwide in 2022, of which 748,000 are in China, ranking third in the world in estimated cases (1). Our current tests for latent tuberculosis infection (LTBI) measure a persistent immune response to antigen stimulation by *Mycobacterium tuberculosis* without tuberculosis-related symptoms (2, 3).

Approximately a quarter of the world population are infected with *Mycobacterium tuberculosis* infection (4, 5). The lifetime risk of developing tuberculosis for persons with *Mycobacterium tuberculosis* infection is estimated to be 5%–10% (6). In order to achieve the WHO End TB strategy by 2035, dealing with the reservoir of *Mycobacterium tuberculosis* infection is essential as it substantially adds to the global tuberculosis burden (5).

**Peer Reviewer** Yu Pang, Beijing Chest Hospital Affiliated to Capital Medical University, Beijing, China

Address correspondence to Qiao Liu, liuqiaonjmu@163.com, or Peijun Tang, tangpeipei001@163.com.

Xinru Fei and Shanshan Wang contributed equally to this article. The order was decided based on the alphabetical order of the authors' last names.

The authors declare no conflict of interest.

See the funding table on p. 11.

The global and regional prevalence of tuberculosis in prisons revealed a prevalence of around 1,173 per 100,000 persons in the Asia-Pacific supervised population, with an estimated global prevalence of approximately 2.8% (7). Detainees bear a disproportionate burden of tuberculosis, facing elevated morbidity and mortality rates compared with the general population (7, 8). National guidance on tuberculosis prevention emphasizes that controlling TB within prisons is pivotal for tuberculosis prevention and control; furthermore, they state that latent infection screening and early intervention play a crucial role in reducing tuberculosis risk. Currently, there are relatively limited and heterogeneous results on LTBI in this special population. A cross-sectional study conducted in an Indian prison revealed a high prevalence of LTBI rate of 64%, while LTBI screening of correctional officers in Brazil identified an infection rate of 23% (9, 10). A study in Qingdao, China, found a LTBI prevalence of 23% among detainees (11). Therefore, it is crucial to understand the current distribution of LTBI and its associated influencing factors in detainees.

Currently, screening methods for LTBI primarily encompass the tuberculin skin test (TST) and interferon-gamma release assay (IGRA), each with its own set of advantages and limitations (12–14). In 2022, the World Health Organization recommended a novel tuberculosis-specific antigen skin test, EC skin test (WHO 2022 Diagnostics Integration Guidelines), which is based on two specific antigens, the ESAT6 and CFP10. Currently, this skin test has shown good discriminative ability in both the general population and tuberculosis patients in China (15), as well as demonstrated promising diagnostic performance in certain special populations (16). However, research on this skin test among the detainees is currently limited. We studied the infection status of new adult female detainees, analyzing risk factors for latent infection in this population and simultaneously evaluating the diagnostic value of the EC skin test overall.

## MATERIALS AND METHODS

### Study subjects

This study systematically evaluated all newly admitted female detainees in a jail in eastern China from 1 October 2022 to 31 October 2023. Detainees were enrolled following rigorous selection criteria. Inclusion criteria included (i) newly recruited detainees in the specified study period; (ii) willing to participate and provide informed consent; (iii) demonstrated capacity to comply with study protocol requirements for latent tuberculosis infection screening procedures. Exclusion criteria: Individuals were excluded from participation if they presented with any of the following conditions, including (i) significant organ dysfunction or diagnosed autoimmune disorders; (ii) prolonged use of immunosuppressive medications, immune modulators, or corticosteroids that could interfere with tuberculosis screening accuracy; (iii) history of tuberculosis infection or current active tuberculosis disease; (iv) known hypersensitivity reactions to tuberculin skin test components or any biological agents employed in the screening protocol.

### Questionnaire

The standardized questionnaire was rigorously developed through a comprehensive literature review and expert consultation. It comprises three primary domains: (i) demographic characteristics (e.g., age, ethnicity); (ii) behavioral and lifestyle factors (e.g., smoking, alcohol consumption); and (iii) medical history (e.g., hypertension, diabetes mellitus). Prior to full deployment, the questionnaire underwent extensive validation through three sequential pilot investigations (total $N = 150$; $n = 50$ per phase). These preliminary assessments confirmed acceptable feasibility and practicality, with participants completing the questionnaire in a mean duration of $12 \pm 3$ min. Evaluation demonstrated robust measurement properties: excellent internal consistency reliability (Cronbach's $\alpha = 0.875$) and strong temporal stability as evidenced by test-retest reliability (intraclass correlation coefficient [ICC] = 0.830; 2-week retest interval).

## Procedures

The structured paper questionnaires were conducted face-to-face by trained professional nurses and then double-checked and recorded into the database by two students. Chest radiographs were taken at the same time of completing the questionnaire survey. Detainees with abnormal chest radiographs need further CT to confirm whether they are patients with active tuberculosis. After active tuberculosis and contraindications have been ruled out, screening for latent tuberculosis infection was conducted. QuantiFERON-TB Gold In-Tube test (QFT-GIT) assay testing was performed on blood samples collected before the EC skin test and TST were carried out, in accordance with the kit manufacturer's instructions.

The TST and EC skin tests were performed on both arms of each detainee at the same time, and the diameters of any abnormal indurations, redness, blisters, or other reactions were documented.

## Skin tests

The TST was manufactured by Chengdu Institute of Biology, China, and administered following the national standard guideline (17). An average diameter of TST induration reaction ≥5 mm as positive, an average diameter of TST induration reaction ≥10 mm as moderate positive, and an average diameter of TST induration reaction ≥15 mm or presence of blisters or other reactions as strongly positive (18). The EC skin test was developed by Zhifei Longcom Biologic Pharmacy. The EC skin test was received on the volar surface of one forearm and TST on the other forearm. The EC antigen is a recombinant reagent of the ESAT-6 and CFP-10 tests, developed by Zhifei Longcom Biologic Pharmacy Company, China. In the EC skin test, the operator drew up 0.1 mL (5 units) of the antigen and administered it into the skin of the volar aspect of the forearm using the Mantoux technique. The millimeter measurements of the transverse and longitudinal diameters of both the induration and redness were taken and recorded between 48 to 72 h post-injection, considering the larger of the mean diameters. A positive EC skin test result was defined as a larger average diameter of induration or redness ≥5 mm, and a presence of blisters or other reactions was defined as strong positive.

## QFT-GIT

Blood samples for the QFT-GIT test were drawn in participants before administering the EC skin test or TST. Then, 6 mL of peripheral blood from the subjects was collected using heparin lithium blood collection tubes and then divided into 1 mL portions in each test tube (QFT-GIT required three test tubes: nil, TB, and mitogen). Samples were cultured at 37℃ for 16–24 h. Post-culturing, the samples were centrifuged for 10 min, and the supernatant was collected and transferred to an enzyme-linked immunosorbent assay (ELISA) microplate. Following a cleaning step, the supernatant was subjected to enzyme labeling for the detection and quantification of IFN-γ concentration. The QFT-GIT (Qiagen, Hilden, Germany) kit and Cellestis Limited's A-QFT software (v2.62) were used for result interpretation per manufacturer's protocol. Results were jugdged according to instructions (19). When Nil ≤8.0 IU/mL, and TB-Nil ≥25% Nil value and ≥0.35, the result is considered positive.

## LTBI

LTBI is defined by any of the following positive diagnostic criteria in the absence of active tuberculosis disease (i) an average induration diameter ≥10 mm or the presence of blisters or other reactions in the TST; (ii) an average diameter of induration or redness ≥5 mm or the presence of blisters or other reactions in the EC skin test; or (iii) a positive QFT-GIT result.

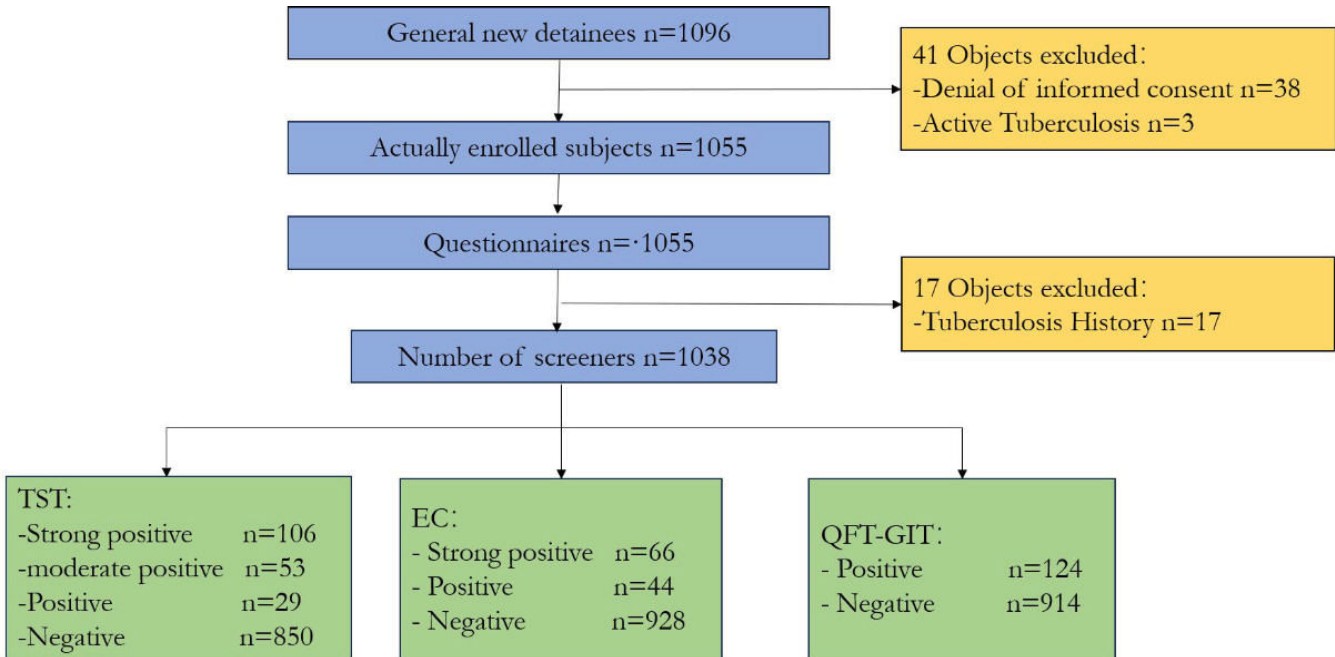

**FIG 1** Flow chart of screening for latent tuberculosis infection in jail detainees.

## Statistical analysis

Statistical analyses were performed using SPSS software (version 25.0; IBM Corporation, Armonk, NY, USA). Categorical variables were expressed as percentages. Intergroup comparisons were analyzed using independent sample $t$-tests and Pearson correlation coefficient analysis. The diagnostic accuracy of the EC skin test was assessed through receiver operating characteristic (ROC) curve analysis, with the area under the curve (AUC) serving as the primary metric of performance. Agreement between dichotomous outcomes from QFT-GIT test, TST, and EC skin tests was evaluated using Cohen's kappa (κ) coefficient, with concordance levels categorized as: poor (κ ≤ 0.20), fair (0.20 < κ ≤ 0.40), moderate (0.40 < κ ≤ 0.60), good (0.60 < κ ≤ 0.80), and excellent (0.80 < κ ≤ 1.00). Statistical significance was defined as $P < 0.05$ for all analyses. For LTBI risk factor assessment, we employed a multivariate logistic regression approach. Variables demonstrating marginal significance ($P < 0.2$) in univariate analyses were subsequently included in multivariable modeling. Four logistic regression models were constructed to evaluate factors associated with LTBI, each using a different definition of a positive LTBI case: (i) positivity on any of the three tests (TST, EC skin test, or QFT-GIT test); (ii) positivity on the QFT-GIT test; (iii) positivity on the EC skin test; and (iv) a TST induration diameter ≥10 mm.

## RESULTS

### Screening test results

From 1 October 2022 to 31 October 2023, a total of 1,096 detainees were newly enrolled, of whom 38 refused informed consent, and three were identified as active tuberculosis after imaging examination, and 17 detainees reported a history of active tuberculosis. Ultimately, 1,038 detainees were enrolled in the study (Fig. 1). In this study, TST, EC, and QFT-GIT screening tests for LTBI were used. For the TST, truncation values for the mean induration diameter were set at 5, 10, and 15 mm. The results indicated that 850 detainees had an induration of less than 5 mm, 29 detainees had indurations between 5 and 10 mm, 53 detainees had indurations between 10 and 15 mm, 106 detainees had indurations of 15 mm or greater or exhibited blistering. For the EC skin test, a cut-off

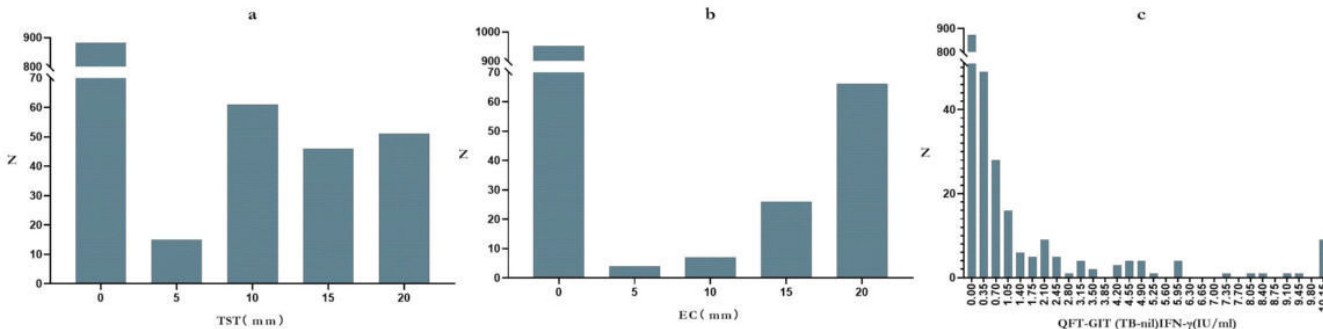

**FIG 2** Distribution results of TST, EC skin test, and QFT-GIT test. (a) TST: X-axis represents average induration diameters (mm); Y-axis indicates the frequency distribution of corresponding diameters. (b) EC: X-axis represents average diameters of induration or redness (mm); Y-axis displays the distribution frequency of corresponding diameters.Detainees with negative TST and EC were labeled as 0 mm combined statistics; the mean diameter of induration or redness ≥ 20mm combined statistics. (c) QFT-GIT: X-axis corresponds to IFN-γ production levels (IU/mL) in TB-nil ; Y-axis displays the distribution frequency of specific IFN-γ concentrations.

value of 5 mm for the mean diameter of induration or redness was applied. The findings showed that 928 detainees had a diameter of <5 mm, 44 detainees measured ≥5 mm, and 66 detainees presented with blisters or other related phenomena. In the QFT-GIT test, 124 detainees were positive, and 914 were negative (Fig. 2). The positive rates of TST, EC skin test, and QFT-GIT were 18.1%, 10.6%, and 11.9% respectively.

## Sensitivity, specificity, and consistency of the EC skin test

Table 1 presents a comprehensive comparative analysis of diagnostic performance between the EC skin test and TST, using the QFT-GIT assay as the reference standard. The results demonstrate markedly superior performance characteristics for the EC skin test compared with the TST. The EC skin test exhibited a sensitivity of 66.9% (95% CI: 58.7%–75.2%) and specificity of 97.0% (95% CI: 95.9%–98.1%), yielding an area under the curve (AUC) of 0.820 (95% CI: 0.778–0.862). In contrast, the TST demonstrated inferior performance with a sensitivity of 46.8% (95% CI: 38.0%–55.6%) and specificity of 85.8% (95% CI: 83.5%–88.0%), corresponding to an AUC of 0.663 (95% CI: 0.617–0.708). The positive predictive value was substantially higher for EC (75.5% vs 30.9%), while both assays maintained comparable negative predictive values (95.6% vs 92.2%). Inter-rater agreement analysis revealed good concordance for EC (κ = 0.673) versus only fair agreement for TST (κ = 0.266). The diagnostic performance of the EC skin test compared with the TST is summarized in Table S1. The AUC for the EC skin test is 0.672 (95% CI: 0.637–0.708), indicating moderate diagnostic accuracy.

**TABLE 1** Diagnostic performance of EC skin test, TST with QFT-GIT test[a]

| Parameter | QFT-GIT | | AUC | Sensitivity | Specificity | PPV | NPV | Kappa | P |
|---|---|---|---|---|---|---|---|---|---|
| | Positive | Negative | (95% CI) | (%, 95% CI) | (%, 95% CI) | (%, 95% CI) | (%, 95% CI) | (95% CI) | |
| TST | | | | | | | | | |
| Positive | 58 | 130 | 0.663 | 46.8 | 85.8 | 30.9 | 92.2 | 0.266 | <0.001 |
| Negative | 66 | 784 | (0.617–0.708) | (38.0–55.6) | (83.5–88.0) | (24.2–37.5) | (90.4–94.0) | (0.191–0.341) | |
| EC | | | | | | | | | |
| Positive | 83 | 27 | 0.820 | 66.9 | 97.0 | 75.5 | 95.6 | 0.673 | <0.001 |
| Negative | 41 | 887 | (0.778–0.862) | (58.7–75.2) | (95.9–98.1) | (67.4–83.5) | (94.3–96.9) | (0.600-0.745) | |

[a]NA, not available; PPV, positive predictive value; NPV, negative predictive value; Kappa coefficients were categorized as poor (κ ≤ 0.20), fair (0.20 < κ ≤ 0.40), moderate (0.40 < κ ≤ 0.60), good (0.60 < κ ≤ 0.80), and very good (0.80 < κ ≤ 1.00).

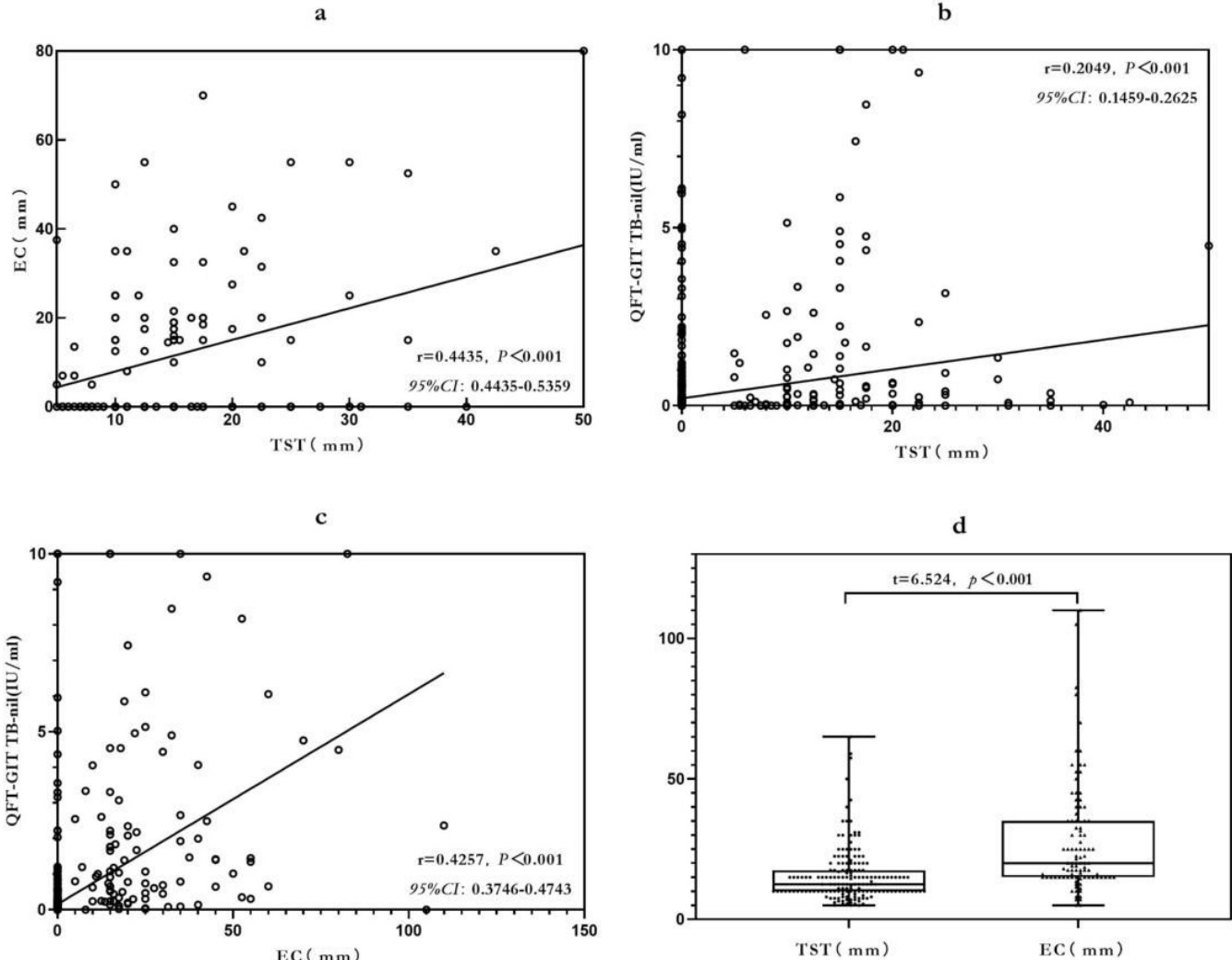

**FIG 3** Correlations and comparisons among TST, EC skin test, and QFT-GIT results. (a) Scatter plot of EC induration (mm) versus TST induration (mm). Each open circle represents an individual participant; the solid line denotes the linear regression fit. (b) Scatter plot of QFT-GIT TB-Nil concentration (IU/mL) versus TST induration (mm). (c) Scatter plot of QFT-GIT TB-Ag concentration (IU/mL) QFT-GIT TB-Nil concentration (IU/mL) versus EC induration (mm). (d) Boxplots comparing distributions of TST and EC induration sizes. Boxes indicate median and interquartile range (IQR) and individual data points are overlaid.

## Diagnostic value of the EC skin test

Positive correlations were observed between EC skin test responses and both TST (r = 0.4435, $P < 0.001$) and QFT-GIT assay results (r = 0.4257, $P < 0.001$). Similarly, TST responses were positively correlated with QFT-GIT results (r = 0.2049, $P < 0.001$). Furthermore, a comparison of the mean induration diameters between the TST and EC skin tests revealed a significantly larger mean diameter for the EC skin test (t = 6.524, $P < 0.001$) (Fig. 3). ROC curves were generated using the categorical QFT-GIT results as the outcome variable and the continuous TST and EC skin test results as predictor variables. The AUC was calculated for each test. The AUC for the TST was 0.663(95% CI: 0.617–0.708), while the AUC for the EC skin test was 0.820 (95% CI: 0.778–0.862), indicating that the EC skin test results had greater concordance with the QFT-GIT results (Fig. 4).

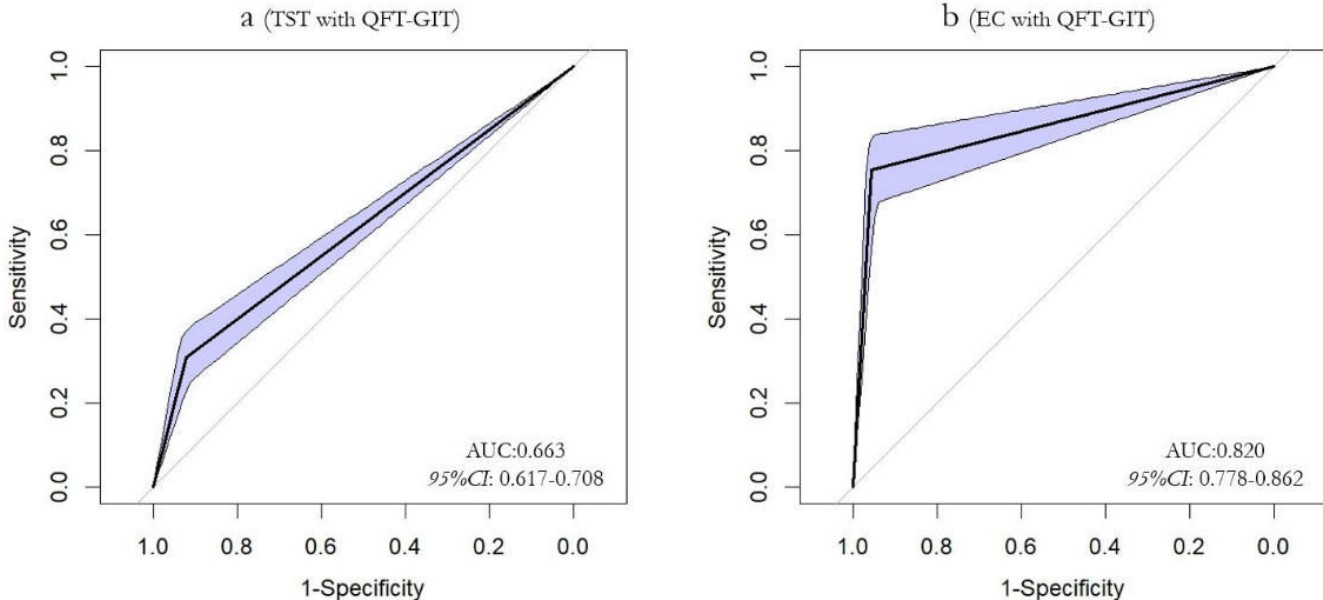

**FIG 4** ROC curve of TST, EC skin test, and QFT-GIT test applied to LTBI screening of detainees. (a) ROC curve of TST and QFT:X-axis represents 1-specificity, and the Y-axis represents sensitivity. (b) ROC curve of EC skin test:X-axis represents 1-specificity, and the Y-axis represents sensitivity.The diagonal is the reference.The shaded area represents the 95% confidence interval.

## Comparisons between LTBI and non-LTBI groups

Of 1,038 detainees, 236 detainees identified as having LTBI through screening tests, while the remaining 802 were classified as non-LTBI. The median age of all detainees was 40 years (interquartile range [IQR]: 32–51), and the median body mass index (BMI) was 23.6 (IQR: 21.5–26.0). Among the detainees, the Han nationality accounted for the highest proportion at 95.8%. Regarding lifestyle habits, 82.1% of the detainees reported no smoking history, 84.6% denied drinking alcohol, and 96.2% denied drug use. Among the 236 LTBI, 6.4% reported current drinking alcohol habits, and 1.7% reported smoking history. Additionally, 6.4% of LTBI reported a history of drug use, and 41.1% reported surgical/traumatic history compared with only 3.0% and 33.5%, respectively, in the non-LTBI group. Significant between-group differences were observed for both drug use history ($\chi^2 = 5.704$, $P = 0.017$) and surgical/traumatic history ($\chi^2 = 4.566$, $P = 0.033$) (Table 2).

## Influencing factors of LTBI

In the model using the combined TST/EC/QFT-GIT criterion, the absence of a history of alcohol consumption (adjusted odds ratio [aOR] = 0.433, 95% CI [CI]: 0.200, 0.938) was identified as a protective factor against LTBI. Conversely, the absence of a history of surgical trauma (aOR = 0.731, 95% CI: 0.539, 0.991) and drug use (aOR = 0.473, 95% CI: 0.233, 0.961) were identified as protective factors against LTBI. In the model using QFT-GIT test positivity as the outcome, the absence of a history of alcohol consumption (aOR = 0.264, 95% CI: 0.104, 0.670) was a protective factor, and minority nationality (aOR = 2.230, 95% CI: 1.042, 4.771) was a risk factor. In the model using TST positivity (≥10 mm) as the outcome, the absence of a history of hepatitis (aOR = 0.431, 95% CI: 0.225, 0.825) was a protective factor. In the model using EC positivity (≥5 mm) as the outcome, the absence of a history of alcohol consumption (aOR = 0.252, 95% CI: 0.101, 0.628) was identified as a protective factor (Table 3).

**TABLE 2** Comparisons between LTBI and non-LTBI group in detention population[a]

| Characteristics | N | Participants | | $\chi^2$/t | P |
|---|---|---|---|---|---|
| | | LTBI | Non-LTBI | | |
| No. of participants | 1,038 | 236 | 802 | | |
| Median age (IQR) | 40 (32–51) | 41 (33–50) | 39 (31–51) | 74.905 | 0.109 |
| Median BMI (IQR) | 23.6 (21.5–26.0) | 23.6 (21.2–26.1) | 23.6 (21.5–26.0) | 444.927 | 0.265 |
| Ethnic | | | | | |
| Han | 994 (95.8) | 224 (94.9) | 770 (96.0) | 0.538 | 0.463 |
| Minority | 44 (4.2) | 12 (5.1) | 32 (4.0) | | |
| Smoke or not | | | | | |
| Never | 852 (82.1) | 188 (79.7) | 664 (82.8) | 4.384 | 0.112 |
| Quit | 156 (15.0) | 44 (18.6) | 112 (14.0) | | |
| Yes | 30 (2.9) | 4 (1.7) | 26 (3.2) | | |
| Drink alcohol or not | | | | | |
| Never | 878 (84.6) | 199 (84.3) | 679 (84.7) | 4.506 | 0.105 |
| Quit | 71 (6.8) | 22 (9.3) | 49 (6.1) | | |
| Yes | 89 (8.6) | 15 (6.4) | 74 (9.2) | | |
| Hypertension history | | | | | |
| Yes | 219 (21.1) | 47 (19.9) | 172 (21.4) | 0.257 | 0.612 |
| No | 819 (87.9) | 189 (80.1) | 630 (78.6) | | |
| Diabetes history | | | | | |
| Yes | 66 (6.4) | 14 (5.9) | 52 (6.5) | 0.093 | 0.760 |
| No | 972 (93.6) | 222 (94.1) | 750 (93.5) | | |
| Drug use or not | | | | | |
| Yes | 39 (3.8) | 15 (6.4) | 24 (3.0) | 5.704 | 0.017 |
| No | 999 (96.2) | 221 (93.6) | 778 (97.0) | | |
| Hepatitis history | | | | | |
| HBV | 29 (2.8) | 4 (1.7) | 25 (3.1) | 4.452 | 0.108 |
| HCV | 115 (11.1) | 19 (8.1) | 96 (12.0) | | |
| No | 894 (86.1) | 213 (90.2) | 681 (84.9) | | |
| Contact history of TB patients | | | | | |
| Yes | 26 (2.5) | 6 (2.5) | 20 (2.5) | 0.002 | 0.966 |
| No | 1,012 (97.5) | 230 (97.5) | 782 (97.5) | | |
| History of surgical trauma | | | | | |
| Yes | 366 (35.3) | 97 (41.1) | 269 (33.5) | 4.566 | 0.033 |
| No | 672 (64.7) | 139 (58.9) | 533 (66.5) | | |

[a]Data indicate medians with interquartile ranges (IQR) or the numbers (%). BMI, body mass index; BCG, Bacillus Calmette-Guérin.

## DISCUSSION

This study aimed to analyze the EC skin test screening outcomes among jail detainees. We found that the EC skin test may have promising potential for widespread implementation in the screening of LTBI.

China is a country where the BCG vaccine is widely administered, which may influence the accuracy of TST results. On the other hand, QFT-GIT is not affected by BCG vaccination and exhibits high specificity (20). Nevertheless, the QFT-GIT test requires laboratory processing and is associated with higher costs. As an alternative, the newly introduced EC skin test shows promise and has the potential to replace TST or QFT-GIT for LTBI screening. The positive rates of TST, EC skin test, and QFT-GIT were 18.1%, 10.6%, and 11.9%, respectively. Our study findings demonstrate a robust agreement between the EC skin test and QFT-GIT screening tests, with high sensitivity and specificity observed for EC. Additionally, the results from Phase 2 clinical trials revealed the safety profile of EC skin test to be favorable during its application (18). Furthermore, a comparative controlled trial demonstrated that the EC skin test exhibited a lower frequency of adverse events compared with TST (21).

**TABLE 3** Multivariable logistic regression analysis of risk factors with latent tuberculosis infection in detainees[a]

| Characteristics | N | TST/EC/QFT-GIT[b] | | | QFT-GIT | | | TST | | | EC | | |
|---|---|---|---|---|---|---|---|---|---|---|---|---|---|
| | | aOR | 95%CI | P | aOR | 95%CI | P | aOR | 95%CI | P | aOR | 95%CI | P |
| Age | 1,038 | 0.989 | 0.977–1.002 | 0.112 | 0.985 | 0.969–1.002 | 0.075 | 1.007 | 0.993–1.022 | 0.343 | 0.989 | 0.972–1.005 | 0.179 |
| BMI | 1,038 | | NA | | | NA | | 0.972 | 0.934–1.011 | 0.158 | | NA | |
| Ethnic | | | | | | | | | | | | | |
| Han | 994 | | NA | | 1 | | | | NA | | | NA | |
| Minority | 44 | | | | 2.230 | 1.042–4.771 | 0.039 | | | | | | |
| Smoke | | | | | | | | | | | | | |
| Yes | 30 | 1 | | | 1 | | | 1 | | | | NA | |
| Quit | 156 | 0.638 | 0.205–1.991 | 0.439 | 1.004 | 0.270–3.732 | 0.995 | 1.964 | 0.572–6.571 | 0.284 | | | |
| No | 852 | 0.471 | 0.151–1.475 | 0.196 | 0.792 | 0.212–2.951 | 0.728 | 2.659 | 0.753–9.393 | 0.129 | | | |
| Drink alcohol | | | | | | | | | | | | | |
| Yes | 89 | 1 | | | 1 | | | | NA | | 1 | | |
| Quit | 71 | 0.649 | 0.341–1.233 | 0.187 | 0.736 | 0.318–1.703 | 0.474 | | | | 0.852 | 0.387–1.875 | 0.691 |
| No | 878 | 0.433 | 0.200–0.938 | 0.034 | 0.264 | 0.104–0.670 | 0.005 | | | | 0.252 | 0.101–0.628 | 0.003 |
| Diabetes history | | | | | | | | | | | | | |
| Yes | 66 | | NA | | | NA | | 1 | | | 1 | | |
| No | 972 | | | | | | | 0.588 | 0.271–1.277 | 0.180 | 1.504 | 0.747–3.030 | 0.253 |
| Hepatitis history | | | | | | | | | | | | | |
| HBV | 29 | 1 | | | 1 | | | 1 | | | 1 | | |
| HCV | 115 | 0.823 | 0.252–2.681 | 0.746 | 3.738 | 0.499–28.014 | 0.199 | 0.662 | 0.225–1.947 | 0.454 | 0.217 | 0.027–1.736 | 0.150 |
| No | 894 | 0.502 | 0.171–1.474 | 0.210 | 0.864 | 0.484–1.544 | 0.621 | 0.431 | 0.225–0.825 | 0.011 | 0.257 | 0.034–1.925 | 0.185 |
| History of surgical trauma | | | | | | | | | | | | | |
| Yes | 366 | 1 | | | 1 | | | 1 | | | 1 | | |
| No | 672 | 0.731 | 0.539–0.991 | 0.043 | 0.692 | 0.468–1.024 | 0.065 | 1.237 | 0.887–1.724 | 0.210 | 0.716 | 0.484–1.059 | 0.094 |
| Drug use | | | | | | | | | | | | | |
| Yes | 39 | 1 | | | 1 | | | 1 | | | 1 | | |
| No | 999 | 0.473 | 0.233–0.961 | 0.038 | 0.593 | 0.249–1.408 | 0.236 | 1.688 | 0.787–3.623 | 0.179 | 0.506 | 0.223–1.148 | 0.103 |

[a]CI, confidence interval; HBV, hepatitis B virus; HCV, hepatitis C virus; NA, not available.
[b]Positivity on any of the three tests (TST ≥10 mm, EC skin test positive, or QFT-GIT test positive).

Although EC sensitivity against QFT GIT was modest, EC demonstrated superior overall diagnostic performance compared with TST, with a higher AUC (0.820 vs 0.663), markedly greater specificity, higher PPV and NPV, and substantially better agreement with QFT GIT (κ = 0.673 vs 0.266). These findings likely reflect the antigen specificity of EC skin test, which reduces cross-reactivity from BCG vaccination and environmental mycobacteria (18). It should be noted that no serological or immunological test is a definitive gold standard for LTBI, and measured sensitivity and specificity are contingent on host- and time-dependent immune responses. Therefore, the modest sensitivity observed does not negate EC potential utility in settings where minimizing false positives is paramount. Future studies with longitudinal follow-up to assess prediction of progression to active TB and comparisons against composite clinical endpoints are warranted to better define the clinical role of EC in LTBI screening.

The selection of screening tests needs to consider various factors, such as policies and economics, so different regions may choose different screening methods or their combinations. Therefore, this study analyzed the risk factors for LTBI of different strategies (TST/EC/QFT-GIT, EC, TST, QFT-GIT) separately. We found that a history of tuberculosis disease is a common risk factor under different strategies. Previous studies have shown that individuals with a history of tuberculosis disease are more likely to experience tuberculosis recurrence or sequelae, such as tuberculosis-associated obstructive pulmonary disease. Additionally, the use of anti-TB drugs and frequent tuberculosis exposure in tuberculosis patients are factors that increase the risk of TB reinfection and morbidity (22). Furthermore, the combined strategy indicates that no

history of drinking alcohol is a protective factor, while surgical trauma history and drug use are the risk factors; the regression analysis results of the QFT-GIT and EC models overlap with the conclusions of the combined application. It has been observed that populations that consume higher levels of alcohol may have higher rates of latent tuberculosis infection among individuals with HIV (23, 24). The subgroup characteristic of surgical trauma has been less involved in previous studies related to LTBI. However, clinical studies have indicated that individuals who have undergone surgery or experienced trauma are more likely to develop varying degrees of complications postoperatively or post-injury and exhibit lowered immune function during a longer recovery period, potentially increasing the risk of *Mycobacterium tuberculosis* infection (25). The results leading to an increased risk of latent tuberculosis infection align with the findings of Nagot et al., where the population with drug use is at least 20 times more likely to develop tuberculosis than the general population (26). Furthermore, the TST model also identified the absence of a history of hepatitis as a protective factor against latent tuberculosis infection. The onset of infectious hepatitis generally accompanies high-risk behaviors (such as blood transfusion, risky sex, etc.) (27, 28), which can also lead to complications like nausea and liver damage. Therefore, under conditions of adverse behavioral exposure and individual immune impairment, the risk of latent tuberculosis infection increases. Previous studies have also indicated that individuals infected with HBV are at higher risk for latent tuberculosis infection (29). Screening for latent pulmonary tuberculosis infection in detainees should prioritize individuals with a history of surgical trauma, tuberculosis disease, drug use, or hepatitis. Additionally, special attention should be paid to cases of long-term alcohol consumption or excessive alcohol abuse among the screening detainees. This targeted screening approach can help identify more individuals with latent tuberculosis infection.

This study was subject to some limitations. First, our sample consisted exclusively of female participants. Given that significant between-sex differences exist in LTBI rates, the female-specific findings cannot be generalized to males. Consequently, future studies should include dedicated male cohort investigations or incorporate sex-stratified analyses in heterogeneous populations. Second, the size of our positive skin test detainees was relatively small. To address this issue, future research could consider enlarging the total sample size of the screening population to increase the number of individuals with positive skin test results.

In conclusion, jail detainees represent a vulnerable population with an elevated risk of tuberculosis, necessitating timely and effective screening for LTBI. The EC skin test demonstrates promising potential as an alternative to traditional diagnostic methods, such as the TST and QFT-GIT assay, for LTBI screening. LTBI screening efforts in this population should prioritize individuals with a history of surgical trauma, prior tuberculosis disease, drug use, or hepatitis. Targeted screening strategies can facilitate the early detection, diagnosis, and management of LTBI.

## ACKNOWLEDGMENTS

We thank for the support from faculties and staffsstaff in study sites.

This study was supported by the Jiangsu Provincial Medical Key Discipline (ZDXK202250), National Nature Science Foundation of China (82003516), and Medical Scientific Research General Project of Jiangsu Health Commission (M2020020).

Qiao Liu and Peijun Tang conceived the study; Xinru Fei, Shanshan Wang, and Qiao Liu analyzed the data and drafted the manuscript; Limei Zhu participated in the study design; Zhan Wang and Xinsong Hu implemented the field investigation; Leonardo Martinez and Cheng Chen participated in the study design and helped draft the manuscript. All authors contributed to the study and have read and approved the final manuscript.

## AUTHOR AFFILIATIONS

[1]Chronic Communicable Disease, Center for Disease Control and Prevention of Jiangsu Province, Nanjing City, Jiangsu Province, China

[2]Department of Epidemiology, Center for Global Health, School of Public Health, Nanjing Medical University, Nanjing City, Jiangsu Province, China

[3]Department of Tuberculosis, The Fourth People's Hospital of Lianyungang, Affiliated hospital of Nanjing Medical University Kangda College, Lianyungang City, Jiangsu Province, China

[4]Department of Epidemiology, School of Public Health, Boston University, Boston, Massachusetts, USA

[5]Department of Pulmonary Disease, The Affiliated Infectious Diseases Hospital of Soochow University, The Fifth People's Hospital of Suzhou, Suzhou City, Jiangsu Province, China

## AUTHOR ORCIDs

Xinru Fei  http://orcid.org/0009-0007-2883-1112
Limei Zhu  https://orcid.org/0000-0002-0448-8547
Peijun Tang  http://orcid.org/0000-0002-8967-8568
Qiao Liu  http://orcid.org/0000-0002-3572-133X

## FUNDING

| Funder | Grant(s) | Author(s) |
|---|---|---|
| National Natural Science Foundation of China | 82574172, 82003516 | Qiao Liu |

## DATA AVAILABILITY

The data dictionary can be made available upon request to the corresponding author.

## ETHICS APPROVAL

This study was reviewed and approved by the ethics committee of the Jiangsu Provincial Center for Disease Control and Prevention (ethical approval number: JSJK2023-B029-02). All eligible participants signed written informed consent.

## ADDITIONAL FILES

The following material is available online.

### Supplemental Material

**Table S1 (Spectrum01500-25-s0001.docx).** Diagnostic performance of the EC skin test and TST.

### Open Peer Review

**PEER REVIEW HISTORY (review-history.pdf).** An accounting of the reviewer comments and feedback.

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
