## [Reviewer comments · Microbiology Spectrum]

Microbiology Spectrum

Diagnostic Accuracy of ESAT6-CFP10 Skin Test for Latent Tuberculosis Infection Among Jail Detainees

Xinru Fei, Shanshan Wang, Zhan Wang, Xinsong Hu, Cheng Chen, Limei Zhu, Leonardo Martinez, Peijun Tang, and Qiao Liu

Corresponding Author(s): Qiao Liu, Nanjing Medical University

Review Timeline:

Submission Date:	May 14, 2025
Editorial Decision:	June 13, 2025
Revision Received:	August 12, 2025
Accepted:	August 20, 2025

Editor: Michael Whitfield

Reviewer(s): Disclosure of reviewer identity is with reference to reviewer comments included in decision letter(s). The following individuals involved in review of your submission have agreed to reveal their identity: Yu Pang (Reviewer #1)

Transaction Report:

DOI: <https://doi.org/10.1128/spectrum.01500-25>

Re: Spectrum01500-25 (**Diagnostic Accuracy of ESAT6-CFP10 Skin Test for Latent Tuberculosis Infection Among Jail Detainees**)

Dear Dr. Xinru Fei:

Thank you for the privilege of reviewing your work. Below you will find my comments, instructions from the Spectrum editorial office, and the reviewer comments.

While we are willing to consider a revised version of this paper at Spectrum, it would be in your best interest to improve the writing. I recommend that you ask a colleague of yours who is a native English speaker to read and provide you some feedback on the writing. You are also welcome to use one of the services here: <https://journals.asm.org/writing-your-paper#language-editing-services>

Revision Guidelines

Sincerely,
Michael Whitfield
Editor
Microbiology Spectrum

Reviewer #1 (Comments for the Author):
Major Comments

The authors investigated the prevalence of latent tuberculosis infection (LTBI) among newly admitted detainees, identified associated risk factors, and evaluated the diagnostic performance of the EC skin test. While the study addresses an important

public health concern, several methodological and analytical issues require substantial revision.

1. The "Methods" section lacks crucial detail regarding participant recruitment. Please elaborate on the sampling strategy, inclusion and exclusion criteria, and the specific procedures used to ensure representative sample selection. Furthermore, a description of the questionnaire design, including its validity and reliability assessments is necessary to support the robustness of the collected demographic and risk factor data.
2. In the "Methods" section, please clearly delineate the positive interpretation criteria for each of the three screening tests (e.g., specific cut-off values for TST and EC skin tests, positive interpretation for IGRA). Crucially, provide a definition of LTBI as classified in this study.
3. The sequencing of the "Results" section, beginning with demographic comparisons between LTBI and non-LTBI groups before presenting the screening test outcomes, is methodologically inverted. The classification into LTBI and non-LTBI groups is inherently dependent on the screening test results. Therefore, the results of the screening tests (e.g., prevalence rates, concordance/discordance) should be presented first to establish group classification, followed by the demographic and risk factor comparisons between these established LTBI and non-LTBI cohorts.
4. For the TST and EC skin tests, a positive cut-off value of 5mm is a widely accepted standard. Figure 2, depicting a group spacing of 2mm (e.g., 0-2mm, 2-4mm), does not align well with standard clinical interpretation thresholds. Following the detailed specification of the skin test judgment criteria as requested should be revised to reflect clinically relevant groupings, such as 0-5mm (negative), 5-10mm (positive/borderline), and >10mm (strongly positive), or similar categories aligned with the study's chosen cut-offs.
5. Given the known high prevalence of various confounding risk factors (e.g., substance use, smoking, comorbidities, nutritional status) in incarcerated populations that can influence both LTBI risk and immune responses, a more robust statistical approach is warranted. We strongly recommend conducting stratified analyses or testing for interaction effects for key demographic and clinical variables to ascertain their independent or modifying impact on LTBI prevalence and diagnostic test performance.
6. Although the study acknowledges the limitation of having only female participants in the conclusion, this critical methodological constraint should be more prominently disclosed in both the Introduction and Methods sections. During the discussion, a more in - depth exploration of the possible impact of this restriction on the results is needed. For example, potential differences between women and men in terms of tuberculosis infection risk, immune response, or test compliance should be considered. Additionally, suggestions on how future research can address this deficiency should be proposed.

Minor Comments

1. The "Methods" section critically lacks the complete manufacturer information for the QuantiFERON-TB Gold In-Tube (QFT-GIT) reagents used in this study. Please specify the manufacturer and country of origin for the QFT-GIT assay.
2. Table 1 presents demographic and clinical characteristics of the LTBI and non-LTBI groups. To effectively illustrate baseline differences or homogeneity between these cohorts, it is essential to include statistical comparisons for all evaluated parameters.
3. Table 2 should be expanded to include the 95% confidence intervals (95% CIs) for the positive predictive value (PPV), negative predictive value (NPV), and Cohen's kappa coefficient.
4. Figure 3 should be supplemented with statistical validation metrics, including correlation coefficients (e.g., Pearson/Spearman r with corresponding P-values), to quantify inter-experimental associations.
5. We recommend adding 95% confidence intervals (95% CIs) to the receiver operating characteristic (ROC) curves in Figure 4.

Reviewer #2 (Comments for the Author):

The authors present a report on the use of the ESAT6-CFP10 skin test for latent tuberculosis screening in newly detained female Chinese prisoners. One major issue with the study is that all patients with a history of tuberculosis infection should have been excluded. Both the skin tests and Quantifeorn assess for an immune response to tuberculosis and they generally do not become negative, even after treatment. For that reason, there is no recommendation to perform LTBI screening on patients with a history of tuberculosis. The study data should be re-evaluated with those patients excluded.

Line 106: Clarify that all detainees are female

Line 115: Spelling of the word Procedures

Line 188: How was conclusive LTBI defined? Was it positive on all 3 LTBI tests? Another metric?

Line 190: BMI is an index. It has no units. Kg/m² is the formula for BMI, not the units.

Line 210: Six Quantiferon results were Indeterminate, not uncertain.

Line 216: A sensitivity for an LTBI screening test of 71.0% is underwhelming. It's possible the EC is more accurate, but that is not discussed anywhere in the manuscript.

Line 223-224: r values of 0.4498 and 0.175 are only moderately correlated. The scatterplots in Figure 3 do not give the impression of a good correlation. That's probably fine, since these tests are used semi-quantitatively and the Quantiferon package insert states the assay shouldn't be used quantitatively. I suggest focusing on the correlation of the categorical results (from lines 216-217) rather than the quantitative results in the discussion (line 273).

Figure 3: Please add a description in the legend for Figure 3D.

Fei *et al.* “Diagnostic Accuracy of ESAT6-CFP10 Skin Test for Latent Tuberculosis Infection Among Jail Detainees” (your reference Spectrum01500-25), Point-by-point response

15 July 2025

Dear Michael Whitfield,

We are pleased to submit a revised version of our manuscript, entitled “*Diagnostic Accuracy of ESAT6-CFP10 Skin Test for Latent Tuberculosis Infection Among Jail Detainees*” (your reference Spectrum01500-25). We thank you and the external reviewers for the careful examination, helpful comments, and the invitation to resubmit a revised manuscript. We reviewed them extensively over a large period of time with all co-authors in order to adequately respond to each Reviewer. We believe we have done so with this revised revision and appreciate the Editor for their patience. We believe the manuscript has been significantly improved in this new version and hope it is ready for publication in your journal.

We have carefully and fully addressed the editorial comments and comments from Reviewer 1 and 2. Please find our point-by-point response, clearly indicating the changes made to the manuscript below.

With this manuscript, we the co-authors would like to confirm that the material has not and will not be offered elsewhere for possible publication while under consideration in your journal. We also confirm that all the contributing authors have reviewed this revision and concur with the revised submission made by the corresponding author.

Please contact the corresponding author (Dr. Qiao Liu) via the above contact details if any further information is needed. We trust that our findings will be of interest to the readership of Microbiol Spectrum.

Sincerely,

Qiao Liu

On behalf of all authors:

Xinru Fei, Shanshan Wang, Zhan Wang, Xinsong Hu, Cheng Chen, Limei Zhu, Leonardo Martinez, Peijun Tang, Qiao Liu

Response to Editor.

Comment #1:

Thank you for the privilege of reviewing your work. Below you will find my comments, instructions from the Spectrum editorial office, and the reviewer comments.

Response:

We thank you for the rapid turn-around of our report, the careful examination, helpful comments, and the invitation to resubmit a revised manuscript. Below, please find a response letter including line-by-line responses to each comment.

Comment #2:

While we are willing to consider a revised version of this paper at Spectrum, it would be in your best interest to improve the writing. I recommend that you ask a colleague of yours who is a native English speaker to read and provide you some feedback on the writing. You are also welcome to use one of the services here: <https://journals.asm.org/writing-your-paper#language-editing-services>

Response:

Thank you for this comment. We have asked Professor Leonardo Martinez who is native English-speaker to revise our paper.

Comment #3:

Response:

Thank you for this comment. We have submitted the revised track changes version to comply with journal publication guidelines.

External Reviewer 1:

Comment #1:

The authors investigated the prevalence of latent tuberculosis infection (LTBI) among newly admitted detainees, identified associated risk factors, and evaluated the diagnostic performance of the EC skin test. While the study addresses an important public health concern, several methodological and analytical issues require substantial revision.

Response:

We wish to thank the Reviewer for the thoughtful review of our manuscript and helpful suggestions to improve this manuscript. We have responded to each point below.

Comment #2:

The "Methods" section lacks crucial detail regarding participant recruitment. Please elaborate on the sampling strategy, inclusion and exclusion criteria, and the specific procedures used to ensure representative sample selection. Furthermore, a description of the questionnaire design, including its validity and reliability assessments is necessary to support the robustness of the collected demographic and risk factor data.

Response:

Thank you for this comment. In response, the 'Study Subjects' section has been revised to provide greater clarity on participant recruitment procedures. Rather than employing sampling methods, this study adopted a facility-wide census approach during the defined study period, enrolling all eligible detainees meeting the refined inclusion/exclusion criteria. These criteria, detailed in section 'Study subjects', were supplemented. We also exclude individuals with prior tuberculosis history, as requested by Reviewer 2. Furthermore, a new subsection titled 'Questionnaire' subsection has been incorporated into the Methods section. This subsection describes the questionnaire's development process and content, and importantly, documents the pre-testing phase and formal assessment of its reliability and validity.

Specifically, we now state,

'Study subjects

This study systematically evaluated all newly admitted female detainees in a jail in eastern China from October 1, 2022 to October 31, 2023. Detainees were enrolled followed rigorous selection criteria. Inclusion criteria included (1) newly recruited detainees the specified study period; (2) willing to participate and provide informed consent; (3) demonstrated capacity to comply with study protocol requirements for latent tuberculosis infection screening procedures. Exclusion criteria: Individuals were excluded from participation if they presented with any of the following conditions included (1) significant organ dysfunction or diagnosed autoimmune disorders; (2) prolonged use of immunosuppressive medications, immune modulators, or corticosteroids that could interfere with tuberculosis screening accuracy; (3) history of tuberculosis infection or current active tuberculosis disease; (4) known hypersensitivity reactions to tuberculin skin test components or any biological agents employed in the screening protocol.

Questionnaire

The standardized questionnaire was rigorously developed through a comprehensive literature review and expert consultation. It comprises three primary domains: (1) demographic characteristics (e.g., age, ethnicity); (2) behavioral and lifestyle factors (e.g., smoking, alcohol consumption); and (3) medical history (e.g., hypertension, diabetes mellitus). Prior to full deployment, the questionnaire underwent extensive validation through three sequential pilot investigations (total N=150; n=50 per phase). These preliminary assessments confirmed acceptable feasibility and practicality, with participants completing the questionnaire in a mean duration of 12±3 minutes. Evaluation demonstrated robust measurement properties: excellent internal consistency reliability (Cronbach's $\alpha = 0.875$) and strong temporal stability as evidenced by test-retest reliability (intraclass correlation coefficient [ICC] = 0.830; 2-week retest interval).'

Lines 105-130

Comment #3:

In the "Methods" section, please clearly delineate the positive interpretation criteria for each of the three screening tests (e.g., specific cut-off values for TST and EC skin tests, positive interpretation for IGRA). Crucially, provide a definition of LTBI as classified in this study.

Response:

Thank you for this comment. The 'Method' section now provides detailed specifications for the interpretative thresholds and positive/negative categorization criteria utilized for both the TST and the EC. Furthermore, the criteria for defining a positive result in the QFT-GIT assay have been explicitly added. Finally, a dedicated subsection has been incorporated to comprehensively outline the operational criteria applied in this study for defining LTBI.

Specifically, we now state,

'An average diameter of TST induration reaction ≥ 5 mm as positive, an average diameter of TST induration reaction ≥ 10 mm as moderate positive, and an average diameter of TST induration reaction ≥ 15 mm or presence of blisters or other reactions as strong positive'

Lines 148-151

'A positive EC skin test result was defined as a larger average diameter of induration or redness ≥ 5 mm, and a presence of blisters or other reactions was defined as strong positive.'

Lines 158-160

LTBI

LTBI is defined by any of the following positive diagnostic criteria in the absence of active tuberculosis disease (1) an average induration diameter of ≥ 10 mm or the presence of blisters or other reactions in the TST; (2) an average diameter of induration or redness ≥ 5 mm or the presence of blisters or other reactions in the EC skin test; or (3) a positive QFT-GIT result.

Lines 175-180

Comment #4:

The sequencing of the "Results" section, beginning with demographic comparisons between LTBI and non-LTBI groups before presenting the screening test outcomes, is methodologically inverted. The classification into LTBI and non-LTBI groups is inherently dependent on the screening test results. Therefore, the results of the screening tests (e.g., prevalence rates, concordance/discordance) should be presented first to establish group classification, followed by the demographic and risk factor comparisons between these established LTBI and non-LTBI cohorts.

Response:

Thank you for this comment. The presentation of the Results section has been comprehensively restructured in response to reviewer' suggestions. We now commence by detailing the distribution of results and positive rates for each screening test. This is followed by the presentation of data assessing the diagnostic value of the EC. The section culminates by grouping participants according to the applied LTBI definition and identifying independent risk factors. Concomitantly, the Conclusions section has been revised to mirror the sequence of the reorganized Results, ensuring logical alignment between the presentation of findings and their interpretation.

Comment #5:

For the TST and EC skin tests, a positive cut-off value of 5mm is a widely accepted standard. Figure 2, depicting a group spacing of 2mm (e.g., 0-2mm, 2-4mm), does not align well with standard clinical interpretation thresholds. Following the detailed specification of the skin test judgment criteria as requested should be revised to reflect clinically relevant groupings, such as 0-5mm (negative), 5-10mm (positive/borderline), and >10mm (strongly positive), or similar categories aligned with the study's chosen cut-offs.

Response:

Thank you for this comment. Table 2 now presents TST and EC results using 5-mm interval groupings. For QFT-GIT analyses, graphical representations were similarly adjusted to reflect the established diagnostic threshold (0.35 IU/mL) for Mycobacterium tuberculosis-specific IFN- γ production (TB-nil), with the x-axis rescaled to highlight this critical cutoff while maintaining appropriate intervals for quantitative TB-nil results. New Figure 2 is stated as below:

Figure 2. Distribution results of TST, EC skin test and QFT-GIT test. (Detainees with negative TST and EC were labeled as 0 mm combined statistics; the mean diameter of induration or redness ≥ 20 mm combined statistics)

Comment #6:

Given the known high prevalence of various confounding risk factors (e.g., substance use, smoking, comorbidities, nutritional status) in incarcerated populations that can influence both LTBI risk and immune responses, a more robust statistical approach is warranted. We strongly recommend conducting stratified analyses or testing for interaction effects for key demographic and clinical variables to ascertain their independent or modifying impact on LTBI prevalence and diagnostic test performance.

Response:

Thank you for this comment. The variable selection strategy for the regression analysis has been refined. Initially, the multivariate model incorporated only variables that achieved a univariate significance level of $P < 0.2$. However, age and BMI have now been included a priori as mandatory covariates in the final multivariate model. This adjustment was made to address the potential confounding effects of these well-established factors, which may differentially influence the risk of LTBI. New Table 3 was stated as below:

Table 3. Multivariable logistic regression analysis of risk factors with latent tuberculosis infection in detainees

Characteristics	N	TST/EC/QFT-GIT*			QFT-GIT			TST			EC			
		aOR	95% CI	P	aOR	95% CI	P	aOR	95% CI	P	aOR	95% CI	P	
Age	1038	0.989	0.977-1.002	0.112	0.985	0.969-1.002	0.075	1.007	0.993-1.022	0.343	0.989	0.972-1.005	0.179	
BMI	1038		NA			NA		0.972	0.934-1.011	0.158		NA		
Ethnic	Han	994		NA		1			NA			NA		
	Minority	44				2.230	1.042-4.771	0.039						
Smoke	Yes	30	1			1		1				NA		
	Quit	156	0.638	0.205-1.991	0.439	1.004	0.270-3.732	0.995	1.964	0.572-6.571	0.284			
	No	852	0.471	0.151-1.475	0.196	0.792	0.212-2.951	0.728	2.659	0.753-9.393	0.129			
Tuberculosis history	Yes	89	1			1			NA		1			
	Quit	71	0.649	0.341-1.233	0.187	0.736	0.318-1.703	0.474			0.852	0.387-1.875	0.691	
	No	878	0.433	0.200-0.938	0.034	0.264	0.104-0.670	0.005			0.252	0.101-0.628	0.003	
Diabetes history	Yes	66		NA		NA		1						
	No	972						0.588	0.271-1.277	0.180	1.504	0.747-3.030	0.253	
Hepatitis history	HBV	29	1			1		1			1			
	HCV	115	0.823	0.252-2.681	0.746	3.738	0.499-28.014	0.199	0.662	0.225-1.947	0.454	0.217	0.027-1.736	0.150
	..	894	0.502	0.171-1.474	0.210	0.864	0.484-1.544	0.621	0.431	0.225-0.825	0.011	0.257	0.034-1.925	0.185
History of surgical trauma	Yes	366	1			1		1			1			
	No	672	0.731	0.539-0.991	0.043	0.692	0.468-1.024	0.065	1.237	0.887-1.724	0.210	0.716	0.484-1.059	0.094
Drug use	Yes	39	1			1		1			1			
	No	999	0.473	0.233-0.961	0.038	0.593	0.249-1.408	0.236	1.688	0.787-3.623	0.179	0.506	0.223-1.148	0.103

Abbreviation: Abbreviation: CI=confidence interval; HBV= Hepatitis B virus; HCV= Hepatitis C virus.NA=not available

* Positivity on any of the three tests (TST \geq 10mm, EC skin test positive, or QFT-GIT test positive).

Comment #7:

Although the study acknowledges the limitation of having only female participants in the conclusion, this critical methodological constraint should be more prominently disclosed in both the Introduction and Methods sections. During the discussion, a more in - depth exploration of the possible impact of this restriction on the results is needed. For example, potential differences between women and men in terms of tuberculosis infection risk, immune response, or test compliance should be considered. Additionally, suggestions on how future research can address this deficiency should be proposed.

Response:

Thank you for this comment. This investigation exclusively focused on a female subpopulation. This limitation is explicitly acknowledged and contextualized within the Introduction, Methods, and Discussion sections.

Specifically, we now state,

‘We studied the infection status of new adult female detainees, analyzing risk factors for latent infection in this population, and simultaneously evaluating the diagnostic value of the EC skin test overall.’

Lines 100-102

‘This study systematically evaluated all newly admitted female detainees in a jail in eastern China from October 1, 2022 to October 31, 2023.’

Lines 107-108

‘First, our sample consisted exclusively of female participants. Given that significant between-sex differences exist in LTBI rates, the female-specific findings cannot be generalized to males. Consequently, future studies should include dedicated male cohort investigations or incorporate sex-stratified analyses in heterogeneous populations.’

Lines 342-346

Comment #8:

The "Methods" section critically lacks the complete manufacturer information for the QuantiFERON-TB Gold In-Tube (QFT-GIT) reagents used in this study. Please specify the manufacturer and country of origin for the QFT-GIT assay.

Response:

Thank you for this comment. The Methods section has been updated to include complete manufacturer details and country of origin for the QFT-GIT assay

Specifically, we now state,

‘The QFT-GIT (Qiagen, Hilden, Germany) kit and Cellestis Limited's A-QFT software (v2.62) were used for result interpretation per manufacturer's protocol.’

Lines 170-172

Comment #9:

Table 1 presents demographic and clinical characteristics of the LTBI and non-LTBI groups. To effectively illustrate baseline differences or homogeneity between these cohorts, it is essential to include statistical comparisons for all evaluated parameters.

Response:

Thank you for this comment. Table 1 has been expanded to include between-group comparisons (LTBI vs. Non-LTBI) of key demographics and baseline characteristics, reporting relevant p-values to demonstrate cohort homogeneity or differences. In accordance with previous comment regarding the sequence of results presentation, Table 1 (original baseline characteristics distribution table) has now been designated as Table 2.

Specifically, we now state,

Table 2. Comparisons between LTBI and Non-LTBI group in detention population.

Characteristics	N	Participants		χ^2/t	P
		LTBI	Non-LTBI		
No. of participants	1038	236	802		
Median age (IQR)	40(32-51)	41 (33-50)	39(31-51)	74.905	0.109
Median BMI (IQR)	23.6(21.5-26.0)	23.6 (21.2-26.1)	23.6(21.5-26.0)	444.927	0.265
Ethnic					
Han	994(95.8)	224(94.9)	770(96.0)	0.538	0.463
Minority	44(4.2)	12(5.1)	32(4.0)		
Smoke or not					
Never	852(82.1)	188(79.7)	664(82.8)	4.384	0.112
Quit	156(15.0)	44(18.6)	112(14.0)		
Yes	30(2.9)	4(1.7)	26(3.2)		
Drink alcohol or not					
Never	878(84.6)	199(84.3)	679(84.7)	4.506	0.105
Quit	71(6.8)	22(9.3)	49(6.1)		
Yes	89(8.6)	15(6.4)	74(9.2)		
Hypertension history					
Yes	219(21.1)	47(19.9)	172(21.4)	0.257	0.612
No	819(87.9)	189(80.1)	630(78.6)		
Diabetes history					
Yes	66(6.4)	14(5.9)	52(6.5)	0.093	0.760
No	972(93.6)	222(94.1)	750(93.5)		
Drug use or not					
Yes	39(3.8)	15(6.4)	24(3.0)	5.704	0.017
No	999(96.2)	221(93.6)	778(97.0)		
Hepatitis history					
HBV	29(2.8)	4(1.7)	25(3.1)	4.452	0.108
HCV	115(11.1)	19(8.1)	96(12/0)		
No	894(86.1)	213(90.2)	681(84.9)		
Contact history of TB patients					
Yes	26(2.5)	6(2.5)	20(2.5)	0.002	0.966
No	1012(97.5)	230(97.5)	782(97.5)		
History of surgical trauma					

Yes	366(35.3)	97(41.1)	269(33.5)	4.566	0.033
No	672(64.7)	139(58.9)	533(66.5)		

*Data indicate medians with interquartile ranges (IQR) or the numbers (%). BMI, body mass index. BCG, bacillus Calmette-Guérin.

Comment #10:

Table 2 should be expanded to include the 95% confidence intervals (95% CIs) for the positive predictive value (PPV), negative predictive value (NPV), and Cohen's kappa coefficient.

Response:

Thank you for this comment. We have added the 95% confidence intervals in Table 1 and Appendix Table1.

Table1. Diagnostic performance of EC skin test, TST with QFT-GIT test.

Parameter	QFT-GIT		AUC (95%CI)	Sensitivity (%, 95%CI)	Specificity (%, 95%CI)	PPV (%, 95%CI)	NPV (%, 95%CI)	Kappa (95%CI)	P
	Positive	Negative							
TST									
Positive	58	130	0.663	46.8	85.8	30.9	92.2	0.266	<0.001
Negative	66	784	(0.617-0.708)	(38.0-55.6)	(83.5-88.0)	(24.2-37.5)	(90.4-94.0)	(0.191-0.341)	
EC									
Positive	83	27	0.820	66.9	97.0	75.5	95.6	0.673	<0.001
Negative	41	887	(0.778-0.862)	(68.7-75.2)	(95.9-98.1)	(67.4-83.5)	(94.3-96.9)	(0.600-0.745)	

* NA= Not Available; PPV: Positive predictive value; NPV: Negative predictive value; Kappa coefficients were categorized as poor ($\kappa \leq 0.20$), fair ($0.20 < \kappa \leq 0.40$), moderate ($0.40 < \kappa \leq 0.60$), good ($0.60 < \kappa \leq 0.80$), and very good ($0.80 < \kappa \leq 1.00$).

Appendix Table 1. Diagnostic performance of EC skin test and TST .

Parameter	TST		AUC (95%CI)	Sensitivity (%, 95%CI)	Specificity (%, 95%CI)	PPV (%, 95%CI)	NPV (%, 95%CI)	Kappa (95%CI)	P
	Positive	Negative							
EC									
Positive	73	37	0.672	38.8	95.6	66.4	87.6	0.411	0.000
Negative	115	813	(0.637-0.708)	(31.9-45.8)	(94.3-97.0)	(57.5-75.2)	(85.5-89.7)	(0.336-0.487)	

* NA= Not Available; PPV: Positive predictive value; NPV: Negative predictive value; Kappa coefficients were categorized as poor ($\kappa \leq 0.20$), fair ($0.20 < \kappa \leq 0.40$), moderate ($0.40 < \kappa \leq 0.60$), good ($0.60 < \kappa \leq 0.80$), and very good ($0.80 < \kappa \leq 1.00$).

Comment #11:

Figure 3 should be supplemented with statistical validation metrics, including correlation coefficients (e.g., Pearson/Spearman r with corresponding P -values), to quantify inter-experimental associations.

Response:

Thank you for this comment. Correlation coefficients supplemented with 95% confidence intervals and respective p -values are now reported for all pairwise comparisons in Figure 3. Additionally, the quantitative intergroup analysis presented in panel d has been augmented with t -statistics and associated p -values.

Figure 3 Scatterplots of the QFT-GIT test, TST, and EC skin test to describe correlation between tests.

Comment #12:

We recommend adding 95% confidence intervals (95% CIs) to the receiver operating characteristic (ROC) curves in Figure 4.

Response:

Thank you for this comment. ROC curves in Figure 4 were regenerated with superimposed 95% confidence interval bands (shaded areas); confidence interval values are now annotated in the bottom-right corner of the plot.

Figure 4. ROC curve of TST, EC skin test and QFT-GIT test applied to LTBI screening of detainees. (a) ROC curve of TST and QFT; (b) ROC curve of EC skin test

External Reviewer 2:

Comment #1:

The authors present a report on the use of the ESAT6-CFP10 skin test for latent tuberculosis screening in newly detained female Chinese prisoners. One major issue with the study is that all patients with a history of tuberculosis infection should have been excluded. Both the skin tests and Quantiferon assess for an immune response to tuberculosis and they generally do not become negative, even after treatment. For that reason, there is no recommendation to perform LTBI screening on patients with a history of tuberculosis. The study data should be re-evaluated with those patients excluded.

Response:

We wish to thank the Reviewer for the thoughtful review of our manuscript and helpful suggestions to improve this manuscript. We have responded to each point below.

We fully acknowledge the reviewers' point regarding the potential impact of a history of tuberculosis (TB) infection on study outcomes, including the possibility of yielding positive results. To address this, individuals with a history of TB disease were excluded during initial screening based on pre-defined exclusion criteria. Furthermore, any participant self-reporting a TB history during questionnaire registration was automatically excluded as ineligible. Consequently, 17 participants reporting a history of TB disease were excluded from the initial pool of 1055 subjects. The final study cohort thus comprised 1038 eligible participants. We have revised all tables and figures in the manuscript based on the latest data from 1038 participants.

Figure. 1 Flow chart of screening for latent tuberculosis infection in jail detainees

‘From October 1, 2022 to October 31, 2023, a total of 1,096 detainees were newly enrolled, of whom 38 refused informed consent and 3 were identified as active tuberculosis after imaging examination and 17 detainees reported a history of active tuberculosis. Ultimately, 1038 detainees were enrolled in the study (Figure 1).’

Lines 209-212

Comment #2:

Line 106: Clarify that all detainees are female

Response:

Thank you for this comment. Line 117 (formerly line 106) explicitly states that the study cohort consisted exclusively of female participants.

Specifically, we now state,

‘This study systematically evaluated all newly admitted female detainees in a jail in eastern China from October 1, 2022 to October 31, 2023.’

Lines 107-108

Comment #3:

Line 115: Spelling of the word Procedures

Response:

Thank you for this comment. We have corrected it.

Specifically, we now state,

‘*Procedures*’

Lines 132

Comment #4:

Line 188: How was conclusive LTBI defined? Was it positive on all 3 LTBI tests? Another metric?

Response:

Thank you for this comment. In this study, latent tuberculosis infection (LTBI) was defined using an integrated interpretation of three screening assays: TST, EC skin test and QFT-GIT. LTBI was diagnosed when any single test yielded a positive result. This diagnostic criterion has been incorporated into the Methods section for clarity.

Specifically, we now state,

LTBI

LTBI is defined by any of the following positive diagnostic criteria in the absence of active tuberculosis disease (1) an average induration diameter of ≥ 10 mm or the presence of blisters or other reactions in the TST; (2) an average diameter of induration or redness ≥ 5 mm or the presence of blisters or other reactions in the EC skin test; or (3) a positive QFT-GIT result.’

Lines 175-180

Comment #5:

Line 190: BMI is an index. It has no units. Kg/m² is the formula for BMI, not the units.

Response:

Thank you for this comment. All body mass index (BMI) unit notations have been removed from the paper.

Specifically, we now state,

‘The median age of all detainees was 40 years (interquartile range (IQR): 32-51), and the median body mass index (BMI) was 23.6 (IQR: 21.5-26.0).’

Lines 256-257

Comment #6:

Line 210: Six Quantiferon results were Indeterminate, not uncertain.

Response:

Thank you for this comment. Among the initial cohort of 1055 participants, six exhibited indeterminate QuantiFERON-TB Gold In-Tube (QFT-GIT) results. Following the exclusion of 17 individuals with prior history of tuberculosis (TB), none of the remaining 1038 participants had indeterminate QFT-GIT results. Consequently, the description of QFT-GIT results in the present manuscript no longer addresses indeterminate outcomes. In future studies, the term "indeterminate" will be rigorously applied exclusively to describe QFT-GIT results according to the manufacturer's specifications.

Specifically, we now state,

'In the QFT-GIT test, 124 detainees were positive and 914 were negative (Figure 2).'

Lines 220

Comment #7:

Line 216: A sensitivity for an LTBI screening test of 71.0% is underwhelming. It's possible the EC is more accurate, but that is not discussed anywhere in the manuscript.

Response:

We appreciate the reviewer's astute observation regarding the diagnostic sensitivity. The results indeed demonstrate that the EC skin test substantially outperforms the TST, exhibiting superior sensitivity, markedly higher specificity, and significantly better overall diagnostic accuracy. We have now added descriptions in the results and discussion sections.

Specifically, we now state,

'Table 1 presents a comprehensive comparative analysis of diagnostic performance between EC skin test and TST, using the QFT-GIT assay as the reference standard. The results demonstrate markedly superior performance characteristics for the EC skin test compared to the TST. The EC skin test exhibited a sensitivity of 66.9% (95% CI: 58.7-75.2%) and specificity of 97.0% (95% CI: 95.9-98.1%), yielding an area under the curve (AUC) of 0.820 (95% CI: 0.778-0.862). In contrast, the TST demonstrated inferior performance with a sensitivity of 46.8% (95% CI: 38.0-55.6%) and specificity of 85.8% (95% CI: 83.5-88.0%), corresponding to an AUC of 0.663 (95% CI: 0.617-0.708). The positive predictive value was substantially higher for EC (75.5% vs 30.9%), while both assays maintained comparable negative predictive values (95.6% vs 92.2%). Inter-rater agreement analysis revealed good concordance for EC ($\kappa=0.673$) versus only fair agreement for TST ($\kappa=0.266$).'

Lines 226-236

'Although EC sensitivity against QFT GIT was modest, EC demonstrated superior overall diagnostic performance compared with TST, with a higher AUC (0.820 vs 0.663), markedly greater specificity, higher PPV and NPV, and substantially better agreement with QFT GIT ($\kappa= 0.673$ vs 0.266). These findings likely reflect the antigen specificity of EC skin test which reduces cross reactivity from BCG vaccination and environmental mycobacteria[19]. It should be noted that no serological or immunological test is a definitive gold standard for LTBI and measured sensitivity and specificity are contingent on host- and time-dependent immune responses. Therefore, the modest sensitivity observed does not negate EC potential utility in settings where minimizing false positives is paramount. Future studies with longitudinal follow up to assess prediction of progression to active TB and comparisons against composite clinical endpoints are warranted to better define the clinical role of EC in LTBI screening.'

Lines 297-307

Comment #8:

Line 223-224: r values of 0.4498 and 0.175 are only moderately correlated. The scatterplots in Figure 3 do not give the impression of a good correlation. That's probably fine, since these tests are used semi-quantitatively and the Quantiferon package insert states the assay shouldn't be used quantitatively. I suggest focusing on the correlation of the categorical results (from lines 216-217) rather than the quantitative results in the discussion (line 273).

Response:

Thank you for this comment. We have restructured our discussion section to emphasize the categorical agreement analyses rather than the quantitative correlations. We have deleted the quantitative correlations part and added more discussion about categorical agreement analyses.

Comment #9:

Figure 3: Please add a description in the legend for Figure 3D.

Response:

Thank you for this comment. We have added the legend for Figure 3A-D.

Specifically, we now state,

‘(a) Scatter plot of EC induration (mm) versus TST induration (mm). Each open circle represents an individual participant; the solid line denotes the linear regression fit. (b) Scatter plot of QFT GIT TB Nil concentration (IU/ml) versus TST induration (mm). (c) Scatter plot of QFT GIT TB Ag concentration (IU/ml) QFT GIT TB Nil concentration (IU/ml) versus EC induration (mm). (d) Boxplots comparing distributions of TST and EC induration sizes. Boxes indicate median and interquartile range (IQR) and individual data points are overlaid.’

Lines 495-500

Figure 3 Correlations and comparisons among TST, EC, and QFT-GIT results.

(a) Scatter plot of EC induration (mm) versus TST induration (mm). Each open circle represents an individual participant; the solid line denotes the linear regression fit. (b) Scatter plot of QFT-GIT TB-Nil concentration (IU/ml) versus TST induration (mm). (c) Scatter plot of QFT-GIT TB-Ag concentration (IU/ml) QFT-GIT TB-Nil concentration (IU/ml) versus EC induration (mm). (d) Boxplots comparing distributions of TST and EC induration sizes. Boxes indicate median and interquartile range (IQR) and individual data points are overlaid.)

Re: Spectrum01500-25R1 (**Diagnostic Accuracy of ESAT6-CFP10 Skin Test for Latent Tuberculosis Infection Among Jail Detainees**)

Dear Dr. Qiao Liu:

Your manuscript has been accepted, and I am forwarding it to the ASM production staff for publication. Your paper will first be checked to make sure all elements meet the technical requirements. ASM staff will contact you if anything needs to be revised before copyediting and production can begin. Otherwise, you will be notified when your proofs are ready to be viewed.

Sincerely,
Michael Whitfield
Editor
Microbiology Spectrum